# Protective Effect of Betulin on Streptozotocin–Nicotinamide-Induced Diabetes in Female Rats

**DOI:** 10.3390/ijms25042166

**Published:** 2024-02-10

**Authors:** Feyisayo O. Adepoju, Ksenia V. Sokolova, Irina F. Gette, Irina G. Danilova, Mikhail V. Tsurkan, Alicia C. Mondragon, Elena G. Kovaleva, Jose Manuel Miranda

**Affiliations:** 1Department of Technology for Organic Synthesis, Institute of Chemical Technology, Ural Federal University, Mira 19, 620002 Yekaterinburg, Russia; besee010@gmail.com (F.O.A.); xenia.socolova@gmail.com (K.V.S.); i.goette@yandex.ru (I.F.G.); ig-danilova@yandex.ru (I.G.D.); 2Institute of Immunology and Physiology, Russian Academy of Sciences, 620049 Yekaterinburg, Russia; 3Leibniz Institute of Polymer Research, 01069 Dresden, Germany; tsurkan@ipfdd.de; 4Departamento de Química Analítica, Nutrición y Bromatología, Campus Terra, Universidade da Santiago de Compostela, 27002 Lugo, Spain; alicia.mondragon@usc.es

**Keywords:** betulin, type 2 diabetes, insulin, islet regeneration, streptozotocin, nicotinamide, antidiabetic

## Abstract

Type 2 diabetes is characterized by hyperglycemia and a relative loss of β–cell function. Our research investigated the antidiabetic potential of betulin, a pentacyclic triterpenoid found primarily in birch bark and, intriguingly, in a few marine organisms. Betulin has been shown to possess diverse biological activities, including antioxidant and antidiabetic activities; however, no studies have fully explored the effects of betulin on the pancreas and pancreatic islets. In this study, we investigated the effect of betulin on streptozotocin–nicotinamide (STZ)-induced diabetes in female Wistar rats. Betulin was prepared as an emulsion, and intragastric treatments were administered at doses of 20 and 50 mg/kg for 28 days. The effect of treatment was assessed by analyzing glucose parameters such as fasting blood glucose, hemoglobin A1C, and glucose tolerance; hepatic and renal biomarkers; lipid peroxidation; antioxidant enzymes; immunohistochemical analysis; and hematological indices. Administration of betulin improved the glycemic response and decreased α–amylase activity in diabetic rats, although insulin levels and homeostatic model assessment for insulin resistance (HOMA–IR) scores remained unchanged. Furthermore, betulin lowered the levels of hepatic biomarkers (aspartate aminotransferase, alanine aminotransferase, and alpha-amylase activities) and renal biomarkers (urea and creatine), in addition to improving glutathione levels and preventing the elevation of lipid peroxidation in diabetic animals. We also found that betulin promoted the regeneration of β–cells in a dose-dependent manner but did not have toxic effects on the pancreas. In conclusion, betulin at a dose of 50 mg/kg exerts a pronounced protective effect against cytolysis, diabetic nephropathy, and damage to the acinar pancreas and may be a potential treatment option for diabetes.

## 1. Introduction

Diabetes mellitus (DM) is a complex metabolic and endocrine disorder characterized by abnormal insulin activity, reduced insulin sensitivity, and hyperglycemia. Diabetes mellitus is a multifactorial illness that is associated with severe secondary consequences, including renal failure, blindness, and early death. According to the International Diabetes Federation, the prevalence of diabetes among adults aged >20–79 is currently 10.5%, and it is projected to increase to 46% by 2045, positioning it as a major worldwide health concern [1].

Type 2 diabetes (T2D) is estimated to be the predominant form of diabetes (90% of all diabetes cases worldwide). It is associated with hyperglycemia due to the progressive decline in insulin-secreting β–cells and a reduction in β–cell volume, together with varying degrees of insulin resistance. Restoring the loss of function of β–cell mass by regeneration is a promising treatment strategy for alleviating some of the burdens of this disease. Hence, compounds that can improve glucose absorption and stimulate the regeneration of insulin-secreting β–cells are promising therapeutic agents.

Preclinical studies modeling diabetes have used streptozotocin and nicotinamide as inducers of T2D in animal models. Streptozotocin induces DNA damage in insulin-secreting cells upon exposure. On the other hand, nicotinamide inhibits the cytotoxic effect of streptozotocin (STZ) through the production of nitric oxide, thereby preventing apoptosis. Rats administered STZ and nicotinamide (NA) exhibit symptoms characteristic of T2D, whereas rats administered only STZ develop type 1 diabetes (T1D) [2]. Despite the progress in the development of numerous therapeutic agents for the treatment of diabetes mellitus in recent years, novel therapeutics are still being sought, aiming at minimal side effects in diabetic patients [2]. Research has demonstrated the efficacy of various bioactive compounds in the management of T2D and associated complications [3,4,5]. One such example is triterpenoids, a diverse class of compounds found in numerous plant species. The antidiabetic activities of these compounds have been studied, and experimental results have demonstrated their ability to inhibit enzymes involved in glucose metabolism and enhance insulin resistance [6,7].

Betulin, a pentacyclic triterpenoid, is the predominant triterpenoid found in birch bark, along with oleanolic acid and betulinic acid. It was also found in several marine organisms, suggesting its much wider presence in nature [8,9]. Betulin is known for its remarkable wound healing ability and other pharmacological advantages. Furthermore, studies have demonstrated that betulin possesses a plethora of pharmacological properties, including anticancer and anti-inflammatory effects, among others [10]. Evidence from studies has shown that betulin can protect the heart against inflammation and oxidative stress caused by ischemia/reperfusion [11] and diabetes [12]. Previous studies have reported the attenuating effect of betulin on high blood glucose, total glycerides, and cholesterol in diabetic rabbits and obese mice [13,14].

However, studies on the antidiabetic effects of betulin, especially regarding the regenerative potential of betulin on β–cells in the islets of Langerhans in T2D diabetic animals, are insufficient. This study evaluated the antidiabetic efficacy of betulin emulsions in streptozotocin–nicotinamide-induced T2D. The antidiabetic parameters were evaluated based on blood glucose parameters, α–amylase activity, liver and kidney function biomarkers, the antioxidant defense system, hematological parameters, and the restoration of β–cell function.

## 2. Results

### 2.1. Quantification of Betulin Content in the Emulsion during the Stability Test

The high-performance liquid chromatography (HPLC) chromatogram produced using the betulin standard under the conditions specified in this investigation (Appendix A) displayed peaks corresponding to betulin at a retention time of 2.03 min. According to the calibration curve (Appendix A), a comparison of the results obtained when determining the mass concentration of betulin before, during, and after the emulsion stability tests indicated that the composition of the betulin emulsion remained stable throughout the stability tests. The obtained data are shown in Appendix A. Visual examination of the emulsion delamination degree during storage revealed no signs of delamination, indicating the durability of the emulsion (Appendix A).

### 2.2. Effect of Betulin on Blood Glucose, Glycosylated Hemoglobin, α–Amylase, and Glucose Tolerance

Fasting blood glucose levels in the diabetic control group were notably greater than those in all the other groups (Appendix A), as illustrated in Figure 1a. Furthermore, treatment with betulin (20 and 50 mg/kg) significantly decreased the blood glucose levels. An increase in plasma glucose accompanied the administration of betulin at a dose of 50 mg/kg to healthy rats, relative to that in the intact group (*p* < 0.05). Nevertheless, we found that betulin had no effect on glycemic control, as the fasting blood glucose (FBG) levels measured in healthy rats treated with water (NC group) and healthy rats treated with betulin (NC + BE50) were not significantly different (*p* > 0.05).

Figure 1b demonstrates that the HbA1c levels were significantly elevated in diabetic controls compared to NC group. Conversely, a dose-dependent reduction in glycosylated hemoglobin (HbA1c) was observed following betulin administration, with a significant difference observed at a dose of 50 mg/kg (the T2D + BE50 group). There was no notable difference in the glycated hemoglobin levels among the healthy rats without any treatment (INT), NC, or NC + BE50 groups (*p* > 0.05).

As illustrated in Figure 1c, betulin treatment decreased the blood α–amylase concentration in diabetic animals in a dose-dependent manner (*p* < 0.05). The administration of betulin at a dose of 50 mg/kg to healthy animals was accompanied by an increase in α–amylase activity compared to that in the INT and NC groups (Figure 1c).

Figure 1d,e show the blood glucose levels of the intact and diabetic groups following the oral delivery of glucose. The elevated blood glucose levels after 120 min and greater area under the curve (AUC) values in diabetic control rats compared to those in the INT group (*p* < 0.05) were evidence of impaired glucose tolerance [15]. The betulin-treated diabetic groups (20 and 50 mg/kg) exhibited improved glucose tolerance, with significantly lower AUC values than the diabetic control group.

### 2.3. Effect of Betulin on the Serum Insulin Concentration and Homeostatic Model Assessment for Insulin Resistance Score

As shown in Figure 1f, there were no significant differences in blood insulin levels among the groups. The homeostatic model assessment for insulin resistance (HOMA-IR) index, a widely employed technique in clinical practice and scientific research for assessing β–cell function and insulin resistance, was computed. Compared to that in the betulin-treated diabetic group and nondiabetic animals, the HOMA-IR index increased marginally in the diabetic control animals (Figure 1g), but the difference was not statistically significant.

### 2.4. Effect of Betulin on Hepatic and Renal Functions in Induced T2D

Several parameters, including creatinine, urea, and protein levels; aminotransferase activity; and alkaline phosphatase activity, were examined in the blood plasma to assess the effects of betulin treatment on the physiological condition of the liver and kidneys. Table 1 shows an increase in the serum aspartate aminotransferase (AST) levels in T2D rats relative to that in the NC group. In contrast, the diabetic groups treated with betulin (T2D + BE20 and T2D + BE50) exhibited lower AST activity than the control group, although the difference was not statistically significant. Furthermore, an increase in alanine aminotransferase (ALT) levels, when compared to those in the normal group, was observed in the T2D group as opposed to the other groups; however, betulin treatment dose-dependently decreased this increase.

In the T2D + BE20 and T2D + BE50 groups, alkaline phosphatase (ALP) activity was reduced not only in comparison to that in the T2D group but also relative to that in the NC group. Concurrently, the administration of betulin (50 mg/kg) to healthy rats was accompanied by a decrease in alkaline phosphatase (ALP) activity relative to that in the INT and NC groups.

Although the total protein level was significantly greater in the T2D + BE50 group than in the T2D + BE20 group, the data presented in Table 1 demonstrate a marked increase in the serum urea and creatinine levels in the T2D group compared to those in the NC group (*p* < 0.05). Nevertheless, administering betulin to diabetic rats resulted in decreased urea levels, from 7.9 ± 0.4 mmol/L in the T2D group to 7.0 ± 0.3 and 5.8 ± 0.1 in the T2D + BE20 and T2D + BE50 groups, respectively. In addition, the administration of betulin to diabetic rats resulted in a reduction in the serum creatinine concentration from 67.7 ± 0.7 µmol/L in the T2D group to 64.3 ± 1.9 µmol/L in the T2D + BE50 group (Table 1).

### 2.5. Effects of Betulin on Oxidative Stress Parameters

The antioxidant response and lipid peroxidation were determined by measuring the levels of reduced glutathione, malondialdehyde (MDA), superoxide dismutase (SOD), and catalase (CAT) in blood plasma. Figure 2a–d demonstrate that the intragastric administration of water to healthy animals in the NC group resulted in a significant increase in SOD activity and glutathione levels in blood plasma compared to those in the INT group (*p* < 0.05). However, neither the MDA levels nor the CAT activity were significantly affected. The administration of betulin to healthy animals (NC + BE50) led to a reduction in glutathione, CAT, and MDA levels in the blood plasma relative to those in the INT and NC groups (Figure 2a–d). The levels of reduced glutathione in the T2D group were lower than those in the healthy rats in the NC group. Moreover, compared with those in the NC group, the plasma concentration of MDA in the diabetic rats was greater, whereas the activities of SOD and catalase were not notably altered.

Following the introduction of betulin at a dose of 20 mg/kg to diabetic rats (T2D + BE20), the glutathione levels in blood plasma increased substantially; however, no significant differences were observed in the levels of MDA, SOD, or CAT when compared to those in the untreated group of rats with T2D. The introduction of a dose of 50 mg/kg betulin to diabetic rats (T2D + BE50) was accompanied by a significant decrease in the concentration of MDA in the blood plasma relative to that in the T2D and T2D + BE20 groups. Furthermore, the concentration of glutathione in the blood plasma of T2D + BE50 rats was lower than that in the blood plasma of NC and T2D + BE20 rats.

### 2.6. Effects of Betulin on Hematological Parameters

The hematological parameters are displayed in Table 2. In comparison to the INT group, the NC group showed significant increases in erythrocyte count, hemoglobin levels, and mean corpuscular hemoglobin concentration (MCHC). Furthermore, no substantial changes were detected in the total number of platelets or leukocytes or in their corresponding fractions, in comparison to those in the INT group. Compared with the INT group, the erythrocyte count, hemoglobin level, and MCHC increased significantly after betulin administration to healthy rats (NC + BE50); however, there was no notable difference compared to those in NC rats (Table 2). In addition, there was no significant alteration in the overall leukocyte, lymphocyte, granulocyte, or mid-range (MID) cell counts in the NC + BE50 group compared to the INT and NC groups. The lymphocyte and MID cell volumes were greater in the T2D group than in the NC group. Additionally, the mean corpuscular hemoglobin (MCH) level in diabetic rats was significantly different from that in the NC group (*p* < 0.05). The leukocyte and lymphocyte counts decreased in diabetic rats treated with betulin (T2D + BE50) compared to the corresponding counts in T2D rats. Moreover, the administration of betulin failed to restore the normal concentration of MCH in diabetic rats (T2D + BE20 and T2D + BE50). No statistically significant differences in platelet counts were observed among the groups listed in Table 2.

### 2.7. Effect of Betulin on Pancreatic Islet β–Cells

Our findings indicate that the total number of islets per field (N/mm^2^) and the percentage of insulin-positive cells in the islets of diabetic rats (T2D group) were lower than those in the NC and INT groups, as indicated by the morphometric and immunohistochemical (IHC) analyses presented in Table 3 and Figure 3. Conversely, the percentage of pancreatic islets with a positive insulin response, the square of the pancreatic islet, the mkm^2^, and the intensity of the cytoplasm in insulin-positive cells did not significantly differ among the groups (Table 3 and Figure 3). In rats with T2D, both treatment doses of betulin (20 and 50 mg/kg) improved the number of pancreatic islets per field (N/mm^2^) and significantly increased the number of insulin+ cells in islets in a dose-dependent manner (Figure 3 and Table 3).

## 3. Discussion

Diabetes is a chronic metabolic disease characterized by hyperglycemia, deficient insulin secretion, or insulin resistance. An experimental T2D model induced by STZ and STZ–NA has been established in previous research [2,16]. In this model, STZ exerts cytotoxic effects on pancreatic β–cells, while nicotinamide partially guards the cells against the action of STZ. In the present study, we evaluated the antidiabetic potential of betulin in regulating the glycemic response, antioxidant enzymes, lipid peroxidation, hepatic and renal parameters, hematological indices, and regeneration of β–cells in T2D animals.

An increase in glucose and glycated hemoglobin levels in the T2D model indicates the occurrence of persistent hyperglycemia in rats. Hyperglycemia, a hallmark of diabetes, is characterized by a fasting blood glucose level above 7.0 mmol/L, an Oral glucose tolerance test (OGTT) above 11.1 mmol/L, or an HbA1c level above 6.5% [1,17]. In diabetic animals, betulin significantly improved the glycemic response in this study by decreasing blood glucose and α–amylase levels, improving glucose tolerance, and normalizing hemoglobin A1c. These findings suggest that betulin may have an antidiabetic effect. During the breakdown of carbohydrates, α–amylase causes an increase in postprandial glucose levels in individuals with diabetes. Inhibiting the activity of this enzyme can delay the breakdown of carbohydrates and diminish postprandial blood glucose levels, ultimately reducing the risk of developing this disease [18]. Our data are consistent with previous studies, in which it was shown that the treatment of diabetic animals with betulin-containing mixtures had a hypoglycemic effect [19,20]. Similarly, others reported that betulin restored insulin resistance by improving glucose tolerance in STZ-induced diabetic rats at 20 and 40 mg/kg doses [21]. The inhibitory effect of betulin on α–amylase in a T2D model has been corroborated by other authors [22].

The biomarkers of hepatic injury include alkaline phosphatase, which measures biliary function, ALT and AST, which measure the amounts of intracellular liver enzymes that have escaped into the bloodstream [23]. Studies have shown that individuals diagnosed with T2D exhibit a greater incidence of abnormalities in hepatic function than individuals without diabetes. Additionally, transient but persistent increases in aminotransferase levels often indicate insulin resistance as an underlying condition [24]. Compared to the control diabetic group, betulin treatment led to a dose-dependent decrease in liver function parameters (ALT, AST, and ALP). The aminotransferase activity of diabetic rodents that were given betulin was restored, which was linked to a decrease in hepatic cytolysis. Furthermore, alkaline phosphatase activity was lower in diabetic rats treated with betulin. These findings showed that betulin influences the activity or velocity of elimination of these enzymes. Previous research has demonstrated reduced AST, ALT, and ALP levels in streptozotocin–nicotinamide-induced diabetic mice after betulinic acid treatment [25].

Elevated urea concentrations in blood plasma can occur because of impaired renal filtration or the utilization of plasma and tissue proteins in gluconeogenesis [26,27]. The accumulation of creatinine in the blood plasma of diabetic rats confirmed that renal filtration was compromised and was indicative of the onset of diabetic nephropathy. Nephropathy in diabetic rats was not extremely pronounced, as there was no observed reduction in plasma protein or total plasma protein. Protein synthesis in the liver was not detected since the total protein concentration remained at the normal level, and the low activity of alkaline phosphatase indicated the absence of cholestatic syndrome [28]. The observed changes in biochemical parameters in the T2D group are predictable for this model, given that the pathogenesis of diabetes mellitus is associated with destructive processes in organs and tissues, including ubiquitous protein glycation, autoimmune processes that target glycated proteins, and oxidative stress [29,30]. The moderate deviation of biochemical parameters from the norm in diabetic rats obtained in our study is a characteristic of T2D, in contrast to T1D [31,32]. Therefore, normalization of the urea and creatinine concentrations at a dose of 50 mg/kg provides confirmation that renal filtration has been restored. These findings align with those of a study conducted by Khataylou et al. [33], which demonstrated that betulinic acid mitigated autoimmune processes in the kidneys of diabetic mice. In addition, betulin attenuates kidney injury by decreasing blood levels of creatinine and urea in a cecal ligation and puncture (CLP)-induced sepsis model in rats [34].

The modeling of T2D within 30 days revealed an imbalance in the antioxidant defense system, due to the accumulation of MDA, a deficiency in glutathione, and the inability to activate antioxidant enzymes. Previous studies have established that T2D is characterized by oxidative stress, as demonstrated by an imbalance in free radical oxidation and the antioxidant defense system [35,36]. Betulin, a triterpenoid, has been shown to exert an antioxidant effect through its ability to scavenge free radicals and increase antioxidant enzymes [37,38,39]. By exerting an antioxidant effect, betulin contributed to a decrease in glutathione consumption and catalase activity in healthy rats treated with betulin at a dose of 50 mg/kg. Furthermore, the restoration of balance in the antioxidant defense system was observed in conjunction with betulin action, as the MDA levels decreased, and the glutathione levels increased, whereas the activity of antioxidant enzymes remained unchanged. Variations in the reduced glutathione concentration are dependent not only on its application as a nonenzymatic antioxidant but also on its production in tissues. It is probable that the T2D + BE50 rats, which were administered a higher dose of betulin, had a lower requirement for GSH than did the T2D + BE20 rats. Additionally, the antioxidant effect of betulin decreased the necessity for SOD and catalase activation in both groups of diabetic animals treated with betulin. Therefore, by facilitating the preservation of glutathione, the antioxidant effect of betulin alleviates the tension in the antioxidant defense (AOD) system. Compared to 20 mg/kg, betulin at a dose of 50 mg/kg is more effective at correcting free radical oxidation and mitigating oxidative stress. Furthermore, Zheng et al. [36] reported that the administration of lycopene, an antioxidant, to diabetic animals resulted in increased plasma antioxidant enzymes, including SOD and catalase. An in vitro-based antioxidant study of betulin by Zhang et al. [40] also demonstrated that betulin exhibits high antioxidant activity.

Elevated hematological indices, including leukocyte, lymphocyte, and platelet counts, have been implicated in diabetic individuals [41]. We found an elevated number of leukocytes and their fractions in diabetic rats compared to healthy rats, which may be associated with increased inflammatory processes in different organs and tissues because of autoimmune aggression against ubiquitously glycated proteins [42,43,44]. Compared to healthy rats, diabetic rats exhibited a greater amount of MCH, which most likely functions as an adaptation to hypoxia, one of the recognized complications of diabetes. In diabetic rodents administered betulin, normalization of the total number of leukocytes and leukocyte fractions indicated a reduction in the inflammatory process. Conversely, an increase in erythrocyte parameters in healthy animals that received intragastric water injections is probably associated with the effect of stress and the release of glucocorticoids during intragastric administration [45,46]. The finding that betulin induces comparable elevations in erythrocyte parameters in healthy animals validates the impact of stress on red blood cell counts. Several leukocytes within the normal range confirmed the absence of an inflammatory process following intragastric injections and betulin treatment. Granulocyte counts within the normal range under the influence of betulin in healthy rats indicate the absence of an acute inflammatory process, while the number of lymphocytes and MID cells, primarily monocytes, indicates the absence of a chronic inflammatory process.

The beneficial effect of betulin on T2D was additionally validated through immunohistochemical and morphological examination of rats induced with streptozotocin–nicotinamide (Appendix A). The regeneration of pancreatic β–cells represent a prospective therapeutic strategy because the pathogenesis of T2D is significantly influenced by the loss of function and bulk of these cells. The administration of betulin to diabetic rats at both doses promoted the regeneration of β–cells. This was corroborated by the observed increases in the quantity of pancreatic islets and insulin-positive cell counts in the T2D + BE20 and T2D + BE50 groups compared to those in the T2D group. Conversely, the lack of substantial variation among groups in terms of the percentage of pancreatic islets with positive insulin staining is considered typical, given that T2D does not result in complete β–cell destruction, and that a subset of β–cells remain viable. Furthermore, no statistically significant differences were observed in the islet area across the different groups. In general, the area of the islet usually increases in individuals with diabetes because of edema, but due to the short duration of the study and the early stages of diabetes (30 days), these increases were not significantly different. However, betulin decreased the islet area in the T2D rats, even though this effect was not statistically significant.

Overall, the protective effect of betulin on pancreatic β–cells and T2D may be attributed to several molecular and chemical targets, including the scavenging of free radicals [10,47,48,49]. Considering that oxidative stress, inflammation, lipid metabolism disorders, and β–cell death are all major contributors to the pathogenesis of T2D and its complications, the antioxidant, anti-inflammatory, and antidyslipidemic properties of betulin, in addition to its ability to promote reparative regeneration, may significantly contribute to its antidiabetic activity. Consistent with our results, several other studies have shown that betulin can significantly inhibit α–amylase production. Studies on the inhibitory effects of different pentacyclic triterpenoids on α–amylase and α–glucosidase enzymes have shown that pentacyclic triterpenoids belonging to the ursane subtype exhibit stronger inhibitory effects against α–amylase and α–glucosidase than other triterpenoids [48]. According to these authors, the diversity of the structural skeleton of pentacyclic triterpenes has a significant impact on the inhibition of α–amylase. Another study showed that betulin had a greater binding affinity for α–amylase (Ki − 13.12 μM; *E_binding_
*− 6.66 kcal/mol) than betulinic acid or curcumin [22]. Because betulin has a stronger inhibitory effect than betulinic acid, researchers have proposed that the C–28 hydroxyl group of betulin might be responsible for its α–amylase inhibition. The authors suggested that the binding of betulin to α–amylase was influenced primarily by hydrogen bonding. Since inhibitors of α–amylase have been studied as an alternative method for the prevention and treatment of T2D, betulin, which is a highly lipophilic compound, may easily cross the membrane and exert its pharmacological effects, such as inhibiting *α*–amylase, thereby delaying the breakdown of carbohydrates and diminishing postprandial blood glucose excursion in people suffering from diabetes.

## 4. Materials and Methods

### 4.1. Preparation of Betulin Emulsion

Betulin obtained from Pro development Ltd. (Yekaterinburg, Russia) was prepared as an emulsion, as described by Zavorokhina et al. [50], with modifications using a high-speed homogenization technique. Betulin was dissolved in a solution of ethanol and glycerol (70:30 at 20 °C) under continuous stirring at 500 rpm and 70 °C for 20 min until thoroughly mixed. The resulting mixture was heated to 40 °C to distill off the ethanol. A betulin solution was introduced dropwise to the external phase containing sodium chloride and an emulsifier, using a flow microreactor (Mr–Lab–VS, Elsoff, Germany) at a feed rate of 0.1 mL/min. The mixture was then heated to 70°C and intensively mixed using an IKA dispersant (T10 basic Ultra–TURRAX, Staufen, Germany).

### 4.2. HPLC Validation of Betulin in Emulsions

Betulin was isolated from the nanoemulsion by employing methylene chloride for extraction, followed by centrifugation and subsequent evaporation of the organic layer to yield a desiccated residue that was reconstituted in HPLC-grade methanol. The betulin content was determined using an Agilent 1290 Infinity chromatographic system (Agilent, Carpinteria, CA, USA) on a reversed stationary phase column (ZORBAX Eclipse Plus C18 RRHD with dimensions of 2.1 × 50 mm and 1.8 µm) with an operating pressure of up to 1200 bar. The mobile phase consisted of a mixture of water and acetonitrile (20:80 (*v*/*v*)) with a flow rate of 0.35 mL/min, an injection volume of 2 µL at 35 °C, a detection wavelength of 210 nm, and a slit width of 4 nm. Identification and quantification of betulin were performed using a calibration curve of 24–56 µg/mL, and 2 µL of betulin was dispensed into the chromatographic system in triplicate. To determine the concentration of betulin in the emulsion, samples packed in vials were enclosed in a constant climate chamber “Binder KBF 240” (Binder, Tuttlingen, Germany) at a temperature of 55 °C. Every 180 h, the vial was removed, cooled, and visually assessed for the degree of emulsion separation, as well as the mass concentration of betulin using HPLC.

### 4.3. Animal Model and Betulin Treatment

The experiment was approved by the ethics committee of the Institute of Immunology and Physiology, Ural Branch of the Russian Academy of Sciences, and carried out in accordance with Directive 2010/63/EU of the European Parliament and the European Council [51]. To induce T2D, overnight fasted rats were administered a single dose of nicotinamide (110 mg/kg, intraperitoneally) 15 min prior to STZ administration (65 mg/kg). Prior to the administration of STZ, the solution was dissolved in 0.1 M citrate buffer (pH 4.5). Streptozotocin and nicotinamide were obtained from Sigma–Aldrich, Saint Louis, MO, USA. Fasting blood glucose levels were estimated, and rats with a blood glucose level ≥7.0 mmol/L were considered diabetic and utilized for the study. The betulin emulsion was orally gavaged at concentrations of 20 and 50 mg/kg, three times a week for 28 days.

### 4.4. Experimental Animals and Design

Female Wistar rats aged 12 weeks and weighing 208 ± 7 g were acquired from the Institute of Immunology and Physiology (UB RAS, Yekaterinburg, Russia). The animals were housed in a polypropylene cage (5 animals per cage) under conditions of optimum light, temperature, and humidity (12:12 h light–dark cycle, 20 ± 2 °C, 50–60%). The rats were fed soy protein-free extruded rodent diets (2020X Teklad, Envigo, Huntingdon, UK) and water ad libitum. Rodents showing no disease symptoms were randomly divided into six groups, each consisting of ten rats (*n* = 10): (1) INT group: healthy rats without any treatment; (2) NC group: healthy rats administered water intragastrically; (3) NC + BE50 group: healthy rats administered betulin emulsion (50 mg/kg body weight, i.g.); (4) T2D group: diabetic control administered water, i.g.; (5) T2D + BE20 group: diabetic rats administered betulin emulsion (20 mg/kg body weight, i.g.); and (6) T2D + BE50: diabetic rats administered betulin emulsion (50 mg/kg body weight, i.g.) for 28 days. Prior studies have shown that betulin at concentrations of 20 and 50 mg/kg can significantly exert hypoglycemic effects on diabetic rats; hence, betulin emulsions were prepared and administered to rats by intragastric injection. [21,52]. All the intervention groups were treated for 28 days, thrice weekly. At the end of the experiment, the rats were deeply anesthetized via an i.m. injection of 0.1 mg/kg b.w. xylazine (Alfasan, Wonderden, The Netherlands) and euthanized via an i.m. injection of 5 mg/kg b.w. Zoletil–100 (Virbac, Carros, France).

### 4.5. Fasting Blood Glucose, Oral Glucose Tolerance Test, and Glycated Hemoglobin

Blood samples were collected from the rats via the tail vein before euthanasia and centrifuged at 1000× *g* for 10 min for biochemical analysis at 4 °C, after which the plasma glucose and glycated hemoglobin (HbA1c) levels were determined using biochemical methods. Plasma glucose levels were determined using the glucose oxidase method with the Novoglyuk–KM Kit, (catalog No 8039; Vector–Best, Novosibirsk, Russian Federation), and glycated hemoglobin (HbA1c) in whole blood was determined via affinity gel chromatography using the Glycohemotest Kit, series No 200497 (ELTA, Moscow, Russian Federation), as described by Danilova et al. [53]. To assess glycemic control in diabetic rats, an OGTT was performed in the intact, T2D, T2D + BE20, and T2D + BE50 groups three days before the end of the experiment. Briefly, the FGB level at 0 min and the postprandial glucose (PG) level measured after oral administration of glucose (1 g/kg) at 30, 60, and 120 min were evaluated. The area under the glycemic curve was calculated between the abscissa and the level of glycemia using the trapezoidal method [54].

### 4.6. Insulin Levels and Homeostatic Model Assessment for Insulin Resistance

The plasma insulin concentration was assayed using an ELISA kit (Cat. No. ERINS; Thermo Fisher Scientific, Waltham, MA, USA). Moreover, insulin resistance was assessed using the homeostasis model assessment (HOMA) approach, as described by Matthews et al. [55]. The HOMA-IR value was determined using the following formula:HOMA-IR = fasting insulin (μU/mL) × fasting glucose (mmol/L)/22.5

### 4.7. Determination of Oxidative Stress Parameters

To evaluate oxidative stress and antioxidant defense parameters, we measured the activity of catalase (EC 1.11.1.6) and superoxide dismutase (SOD; EC 1.15.1.1) in erythrocyte hemolysates, as well as the plasma levels of thiols, including reduced glutathione. The content of TBA-reactive products, including MDA, was also determined spectrophotometrically. Erythrocytes were hemolyzed in distilled water and centrifuged at 1000× *g* for 10 min.

Catalase activity in the hemolysates was determined by measuring the loss of hydrogen peroxide, which can form a colored complex with ammonium molybdate [56]. Three ml of 0.1 M phosphate buffer (pH 7.4) containing 1% hydrogen peroxide and 10 μL of hemolysate was incubated for 10 min at 37 °C, after which the reaction was stopped by the addition of ammonium molybdate. To confirm that the concentration of the hydrogen peroxide solution was 0.3%, the solution was titrated against potassium permanganate and sulfuric acid (standard titers from EKROSHIN LLC, Moscow, Russia). SOD activity was determined by the inhibition of the diformazan formation from nitroblue tetrazolium in the presence of riboflavin and methionine [57].

The determination of thiols and reduced glutathione was based on the reaction of –SH groups with 5,5′–dithiobis(2–nitrobenzoic) acid (DTNB; Abcam, Cambridge, UK) or Ellman’s reagent at pH 8.0 [58]. For this assay, the supernatant obtained after the precipitation of plasma proteins with 5.7% trichloroacetic acid in 0.25 M HCl (JSC Vekton, Moscow, Russia) was centrifuged at 1000× *g* for 10 min.

Malondialdehyde is a byproduct of free radical oxidation (FRO) that occurs after the formation of earlier products, hydroperoxides and diene conjugates, which are subsequently converted to MDA and other aldehydes. Malondialdehyde and other FRO products containing aldehyde groups form compounds with thiobarbituric acid (TBA; Diaem, Moscow, Russia) in an acidic environment when heated. The optical density measured at a wavelength of 532 nm is directly proportional to the MDA concentration [59].

### 4.8. Hepatic and Renal Function Parameters

AST and ALT enter the bloodstream when the integrity of the membrane is compromised, so AST and ALT activity are traditionally used to assess damaging processes in the liver, myocardium, and other tissues. An elevated AST/ALT ratio indicates prevalent myocardial injury, whereas a reduced AST/ALT ratio indicates liver injury.

Alpha–amylase activity increases in blood plasma when the exocrine pancreas is damaged, and this enzyme is also a target for antidiabetic compounds, since a decrease in its activity helps reduce glucose consumption [60]. In diabetic nephropathy, there is an elevation in the levels of urea and creatinine, while the total protein levels decrease [61].

The plasma levels of AST (catalog No B 00.102), ALT (catalog No B 00.102), ALP (catalog No B 09.02), total protein (catalog No B 06.01), urea (B 08,02), α–amylase (catalog No B 11.01), and creatinine (catalog No B 04.02) were evaluated using kits provided by Vital Diagnostics (St. Petersburg, Russia). Subsequently, a spectrophotometer (DU–800 Beckman Coulter Int. S.A., Nyon, Switzerland) was used to measure the absorbance at a specified wavelength, following the instructions provided by the manufacturers.

### 4.9. Hematological Parameter Determination

Hematological parameters such as white blood cell (WBC) count, lymphocyte count, MID cell count, erythrocyte count, hematocrit, hemoglobin, granulocyte count, MCH, MCHC, and platelet (PLT) count were estimated using an automated hematology analyzer (Biocode Hycel, Paris, France).

### 4.10. Immunohistochemical Evaluation of Pancreatic Tissues

A median laparotomy was conducted to retrieve the pancreas. Pancreatic tissues were excised and fixed in a 10% formalin solution for 24 h at room temperature. The tissues were then prepared for standard histological analysis and embedded in paraffin blocks using an automated processor (Leica EG 1160, Illinois, IL, USA). Tissue sections with a thickness of 3–4 mm were prepared using a Leica SM 2000R sliding microtome (Illinois, IL, USA). An immunohistochemical procedure was performed using the avidin–biotin peroxidase complex (ABC) method. To detect insulin-positive cells, 4 μm thick pancreas paraffin sections were deparaffinized, dehydrated, and washed in phosphate-buffered saline (3×; pH 6.0) at room temperature for 5 min. The pancreatic tissues were subjected to overnight incubation at 4 °C with anti-insulin and pro-insulin + insulin antibodies (clone INS04 + INS05, MA5–12042; Invitrogen, Carlsbad, CA, USA), diluted at 1:200. Following incubation with a biotinylated secondary antibody for 1 h at 37 °C, a diaminobenzidine (DAB)–nickel reaction was conducted, and all tissue sections were counterstained using hematoxylin and viewed under a microscope (Leica DM 2500, Illinois, IL, USA) at an objective magnification of 40×.

### 4.11. Morphometric Analysis

Using pancreatic tissue slides stained with insulin-conjugated IHC antibodies, insulin-positive cells were detected. The overall number of islets and the number of islets exhibiting a positive response to insulin in 1 mm^2^ of the pancreatic parenchyma (N/mm^2^) were also recorded. The percentage of insulin-positive cells in each individual islet was calculated by dividing the total number of islet cells by the number of cells that displayed positive insulin staining (Appendix A). The insulin content in β–cells was evaluated by measuring the intensity of insulin antigen expression in the islets according to their optical density, using the image analysis program Video Test “Morphology” 5.0 (VideoTesT, St. Petersburg, Russia).

### 4.12. Statistical Analysis

The analysis was performed using Microsoft Office Excel 365 and Origin Pro 9.0 software. The data are presented as the mean ± standard deviation (SEM), and the significance level was set at *p* < 0.05. The Mann–Whitney U test was used to analyze the nonparametric independent observations.

## 5. Conclusions

In our study, T2D was accompanied not only by persistent hyperglycemia but also by a change in biochemical parameters suggesting damage to the tissue of the pancreas, liver, and kidneys; a change in hematological parameters indicating the occurrence of an inflammatory process and hypoxia; and oxidative stress confirmed by an increase in the activity of antioxidant enzymes that do not prevent the accumulation of malondialdehyde in the blood. Intragastric administration of betulin at doses of 20 mg/kg and 50 mg/kg to diabetic rats contributed to the normalization of glycated hemoglobin and a decrease in glucose levels and biochemical parameters characterizing damage to the pancreas, kidneys, and liver. Furthermore, the administration of betulin to diabetic rats had an antioxidant effect and counteracted oxidative stress, as well as increasing the area of pancreatic islets and the percentage of insulin-positive cells in the pancreatic islets of the treated diabetic groups.

Additionally, our study underscores the social significance of betulin as a potential therapeutic agent for diabetes. Notably, betulin is easily extractable and devoid of patent restrictions, making it an accessible and cost-effective compound. This characteristic holds immense promise for underserved and economically challenged regions where the availability of expensive diabetic drugs is limited.

The limitations of our study mostly stem from its narrow focus on the effects of betulin treatment in the early stages of experimental diabetes (30 days). We did not perform pharmacokinetic studies when discussing the pharmacodynamics of the treatment, nor did we specify betulin’s mechanism of action or molecular targets. Finally, we demonstrated the ability of betulin to reduce hyperglycemia and oxidative stress without causing toxicity; however, we did not explore the effects of betulin therapy on lipid metabolism or inflammation, both of which play a role in the pathogenesis of diabetes.

## Figures and Tables

**Figure 1 ijms-25-02166-f001:**
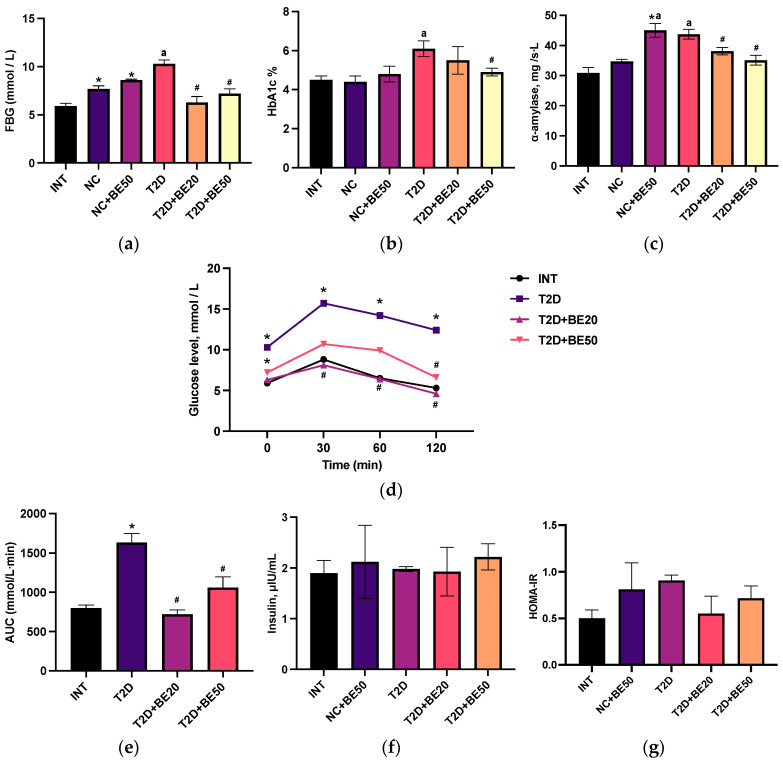
(**a**–**g**) Effects of betulin treatment on healthy and diabetic rats: (**a**) fasting blood glucose, (**b**) glycosylated hemoglobin, (**c**) α–amylase, (**d**) oral glucose tolerance test, (**e**) area under the curve, (**f**) insulin, and (**g**) the HOMA-IR index. The values (**a**,**c**,**f**) were measured in plasma, whereas (**b**,**d**) were measured in the whole blood. INT—healthy rats without any treatment, NC—healthy rats administered water, NC + BE50—healthy rats administered betulin (50 mg/kg), T2D—diabetic rats administered water, T2D + BE20—diabetic rats administered betulin (20 mg/kg), T2D + BE50—diabetic rats administered betulin (50 mg/kg). *N* = 10 in each group. The data are presented as the mean ± SEM. Significant differences between groups were determined through the Mann–Whitney U test: *—indicates a significant difference in comparison to INT (*p* < 0.05); #—indicates a significant difference in comparison to T2D (*p* < 0.05); ^a^—indicates a significant difference in comparison to NC (*p* < 0.05).

**Figure 2 ijms-25-02166-f002:**
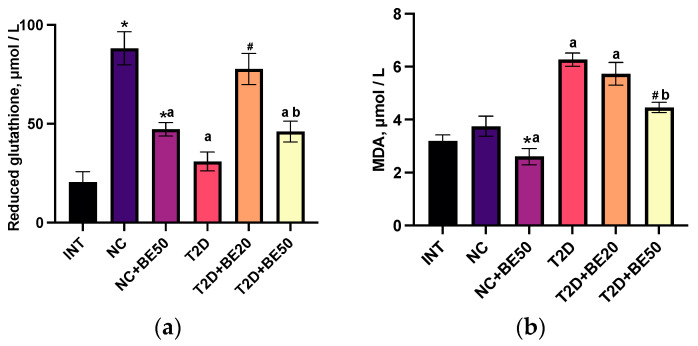
(**a**–**d**) Effects of betulin treatment on the plasma levels of (**a**) reduced glutathione and (**b**) malondialdehyde (MDA), and activity of (**c**) superoxide dismutase (SOD) and (**d**) catalase (CAT) in erythrocyte hemolysates. INT—healthy rats without any treatment, NC—healthy rats administered water, NC + BE50—healthy rats administered betulin (50 mg/kg), T2D—diabetic rats administered water, T2D + BE20—diabetic rats administered betulin (20 mg/kg), T2D + BE50—diabetic rats administered betulin (50 mg/kg). *N* = 10 in each group. The data are presented as the mean ± SEM. Significant differences between groups were determined through the Mann–Whitney U test: *—indicates a significant difference in comparison to INT (*p* < 0.05); ^#^—indicates a significant difference in comparison to T2D (*p* < 0.05); ^a^—indicates a significant difference in comparison to NC (*p* < 0.05); ^b^—indicates a significant difference in comparison to T2D + BE20 (*p* < 0.05).

**Figure 3 ijms-25-02166-f003:**
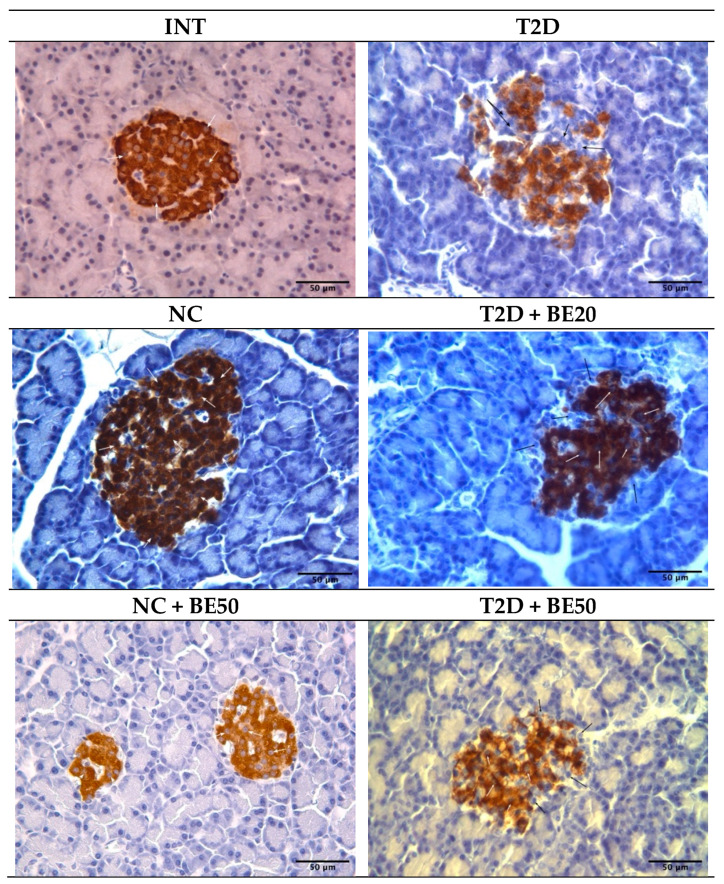
Photomicrograph of pancreatic sections showing significant insulin positive cells (brown stain–white arrow) and insulin-negative cells (black arrow) at ×400 magnification. INT–healthy rats without any treatment, NC—healthy rats administered water, NC + BE50—healthy rats administered betulin (50 mg/kg), T2D—diabetic rats administered water, T2D + BE20—diabetic rats administered betulin (20 mg/kg), T2D + BE50—diabetic rats administered betulin (50 mg/kg).

**Table 1 ijms-25-02166-t001:** Effect of betulin intervention on hepatic and renal functions.

	Aspartate Aminotransferase µmol/min∙L	Alanine Aminotransferase µmol/min∙L	Alkaline Phosphatase µmol/min∙L	Total Protein g/L	Urea mmol/L	Creatinine µmol/L
INT	16.1 ± 0.6	12.6 ± 0.6	51.5 ± 3.3	70.5 ± 1.3	5.2 ± 0.2	62.3 ± 1.5
NC	18.5 ± 0.8	12.8 ± 0.7	48.5 ± 4.3	67.3 ± 2.2	5.9 ± 0.3	60.2 ± 1.7
NC + BE50	17.4 ± 1.7	13.1 ± 0.8	32.0 ± 2.4 *^,a^	67.6 ± 1.3	5.2 ± 0.4	64.3 ± 1.6
T2D	24.7 ± 1.7 ^a^	18.6 ± 1.2 ^a^	40.8 ± 4.0	64.3 ± 2.1	7.9 ± 0.4 ^a^	67.7 ± 0.7 ^a^
T2D + BE20	21.7 ± 1.4	13.6 ± 0.8 ^#^	30.4 ± 1.4 ^a,#^	62.1 ± 1.6	7.0 ± 0.3 ^a^	68.0 ± 3.3 ^a^
T2D + BE50	20.2 ± 2.6	11.3 ± 1.4 ^#^	27.6 ± 1.7 ^a,#^	68.0 ± 1.5 ^b^	5.8 ± 0.1 ^#,b^	64.3 ± 1.9

Concentrations in plasma. The data are presented as the mean ± SEM. Significant differences between groups were determined through the Mann–Whitney U test: *—indicates a significant difference in comparison to healthy rats without any treatment (INT) (*p* < 0.05); ^#^—indicates a significant difference in comparison to T2D (*p* < 0.05); ^a^—indicates a significant difference in comparison to healthy rats administered water (NC) (*p* < 0.05); ^b^—indicates a significant difference in comparison to T2D + BE20 (*p* < 0.05). NC + BE50—healthy rats administered betulin (50 mg/kg), T2D—diabetic rats administered water, T2D + BE20—diabetic rats administered betulin (20 mg/kg), T2D + BE50—diabetic rats administered betulin (50 mg/kg). *N* = 10 in each group.

**Table 2 ijms-25-02166-t002:** Effect of betulin intervention on hematological parameters in whole blood.

Parameter	INT	NC	NC + BE50	T2D	T2D + BE20	T2D + BE50
Leukocytes (×10^3^/µL)	7.73 ± 0.41	7.71 ± 0.43	6.18 ± 0.57	7.61 ± 0.56	7.98 ± 0.83	6.74 ± 0.58
Lymphocytes (×10^3^/µL)	5.10 ± 0.40	5.14 ± 0.48	4.18 ± 0.46	5.65 ± 0.48	6.16 ± 0.44	4.98 ± 0.52
Mid-range cells (×10^3^/µL)	0.72 ± 0.18	0.74 ± 0.10	0.45 ± 0.09	0.95 ± 0.16	0.84 ± 0.13 ^#^	1.04 ± 0.10 ^#^
Granulocytes (×10^3^/µL)	1.92 ± 0.42	1.53 ± 0.32	1.62 ± 0.24	1.63 ± 0.39	0.98 ± 0.38	0.72 ± 0.14
Erythrocytes (×10^6^/µL)	8.01 ± 0.28	9.14 ± 0.22 *	9.06 ± 0.20 *	8.88 ± 0.32	8.54 ± 0.44	9.16 ± 0.27
Hemoglobin (g/L)	13.7 ± 0.4	16.2 ± 0.8 *	15.1 ± 0.2 *	16.2 ± 0.3	16.1 ± 0.7	17.2 ± 0.5
Hematocrit (%)	41.2 ± 1.2	43.4 ± 0.3	42.9 ± 0.7	43.8 ± 0.7	47.4 ± 2.7	45.7 ± 1.6
MCH (pg)	17.1 ± 0.3	17.5 ± 0.4	16.7 ± 0.4	18.7 ± 0.2 ^#^	18.9 ± 0.2 ^a^	18.7 ± 0.1 ^a^
MCHC (g/dL)	33.2 ± 0.5	35.8 ± 0.8 *	35.3 ± 0.3 *	36.6 ± 0.6	34.4 ± 0.7	37.6 ± 0.3 ^b^
PLT (×10^3^/µL)	602.0 ± 26.7	487.7 ± 59.8	578.8 ± 57.8	565.8 ± 20.1	432.4 ± 65.7	529.0 ± 14.7

MCH—mean corpuscular hemoglobin, MCHC–mean corpuscular hemoglobin concentration, PLT—platelet. INT—healthy rats without any treatment, NC—healthy rats administered water, NC + BE50—healthy rats administered betulin (50 mg/kg), T2D—diabetic rats administered water, T2D + BE20—diabetic rats administered betulin (20 mg/kg), T2D + BE50—diabetic rats administered betulin (50 mg/kg). *N* = 10 in each group. The data are presented as the mean ± SEM. Significant differences between groups were determined through the Mann–Whitney U test. *—indicates a significant difference in comparison to INT (*p* < 0.05); ^#^—indicates a significant difference in comparison to T2D (*p* < 0.05); ^a^—indicates a significant difference in comparison to NC (*p* < 0.05); ^b^—indicates a significant difference in comparison to T2D + BE20 (*p* < 0.05).

**Table 3 ijms-25-02166-t003:** Morphometric analysis of the pancreatic islets of rats.

Parameter	INT	NC	NC + BE50	T2D	T2D + BE20	T2D + BE50
% of pancreatic islets with positive insulin staining	100.00 ± 0.00	99.55 ± 0.45	100.00 ± 0.00	98.10 ± 1.31	93.53 ± 5.49	100.00 ± 0.00
Number of pancreatic islets, N/mm^2^	4.19 ± 0.56	3.18 ± 0.24	2.38± 0.12 ^#,a^	1.53 ± 0.12 *^,a^	2.45 ± 0.41 *	3.34 ± 0.47 ^#^
Area of pancreatic islet, mkm^2^	7426 ± 1331	9995 ± 726	7333.51 ± 1922	9083 ± 919	6699 ± 1153	6990.55 ± 1231
% of IPC in the pancreatic islet	79.71 ± 1.37 ^a^	85.63 ± 1.90 ^#^	93.77 ± 1.23 *^,b,#^	57.22 ± 7.97 *^,a^	65.83 ± 11.35	78.80 ± 2.11 ^#^
Optical intensity of cytoplasm in IPC	0.41 ± 0.03	0.63 ± 0.04	0.51 ± 0.07	0.47 ± 0.06	0.61 ± 0.03	0.54 ± 0.07

IPC—insulin-producing cells. INT—healthy rats without any treatment, NC—healthy rats administered water, NC + BE50—healthy rats administered betulin (50 mg/kg), T2D—diabetic rats administered water, T2D + BE20—diabetic rats administered betulin (20 mg/kg), T2D + BE50—diabetic rats administered betulin (50 mg/kg). *N* = 10 in each group. The data are presented as the mean ± SEM. Significant differences between groups were determined through the Mann–Whitney U test: *—indicates a significant difference in comparison to INT (*p*< 0.05); ^#^—indicates a significant difference in comparison to T2D (*p* < 0.05); ^a^—indicates a significant difference in comparison to NC (*p* < 0.05); ^b^—indicates a significant difference in comparison to T2D + BE20 (*p* < 0.05).

## Data Availability

Data are contained within the article and Appendix A.

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
