# Peer review of "Protective Effect of Betulin on Streptozotocin–Nicotinamide-Induced Diabetes in Female Rats"

_ijms, 2024, doi:10.3390/ijms25042166_

Round 1

Reviewer 1 Report

Comments and Suggestions for Authors

This study examined the anti-diabetic effects of the natural product betulin in an animal model of streptozotocin nicotinamide induced hyperglycemia.  The authors show betulin having a significant effect of reducing fasting blood glucose  and oral glucose tolerance test without increasing indicators for renal or hepatic toxicity with data indicating that betulin alters pancreatic islet morphology and quantity of insulin positive cells.

 Several significant issues must be addressed prior to publication.  See below for a list of concerns.

Lines 100 and 105 indicate no significant difference and that the p- value equaled 0.05 for the statistical test used.  Is 0.05 the actual value or was the p-value more than 0.05?  Most studies use a p value of ≤ 0.05 as the cut off for stating statistically significant differences exist.  If the p-value was actually 0.05 it may be argued that there are significant differences between treatment groups. 

For all table and figure legends; please include 1) N value per treatment group 2) the biological sample (e.g. blood, plasma, etc…) used to collect the data and 3) the statistical test used to determine significant differences. 

Figure 2a shows that oral gavage or intragastric (NC) results in a statistically significant and four fold increase in glutathione.  How does oral dosing of water increase reduced glutathione?  The authors need to address this question. The authors should also specify that it is reduced glutathione not oxidized in the figure legend or Y axis label in Figure 2.  

The counterintuitive aspect of diabetic model data presented in Table 4 and fasting blood insulin (Figure 1f) need to be addressed.  Specifically, Table 4shows T2D group having significantly less number if islets per surface area (2nd row down) and percent insulin producing cells (4th row down) while Figure 1f shows no change in fasting blood insulin.  Why isn’t there significant differences in fasting insulin values if BOTH the number of islets and the percent of insulin positive cells per islet are less?  The authors need to address this question.

Author Response

With respect to the comments from the Reviewer â„– 1:

Point 1:
Lines 100 and 105 indicate no significant difference and that the p- value equaled 0.05 for the statistical test used. Is 0.05 the actual value or was the p-value more than 0.05? Most studies use a p value of ≤ 0.05 as the cut off for stating statistically significant differences exist. If the p-value was actually 0.05 it may be argued that there are significant differences between treatment groups.

Response 1: Thank you very much for your comment. In fact, the cutoff point used to consider a statistically significant results was P < 0.05 (and not equaled 0.05). It is a classical approach, although we understand its limitations [https://www.ncbi.nlm.nih.gov/pmc/articles/PMC5750014; https://onlinelibrary.wiley.com/doi/10.1111/brv.12315]. Concerning the value cited in lines 100 and 105 (p = 0.05) it was a mistake, we should have put p<0.05 for the cases with no significant differences were found, and ≥0.05 in the cases that not significant differences were found. We have corrected this concern throughout the revised version of the manuscript.

Point 2:
For all table and figure legends; please include 1) N value per treatment group 2) the biological sample (e.g., blood, plasma, etc…) used to collect the data and 3) the statistical test used to determine significant differences.

Response 2: According to the suggestions from the Reviewer, all the required information (N values, biological sample and statistical test used) were added to all the table and figure legends.

Point 3:
Figure 2a shows that oral gavage or intragastric (NC) results in a statistically significant and four-fold increase in glutathione. How does oral dosing of water increase reduced glutathione? The authors need to address this question. The authors should also specify that it is reduced glutathione not oxidized in the figure legend or Y axis label in Figure 2.

Response 3: Oral dosing of water is a stress factor. Reduced glutathione is a major antioxidant in mammalian cells, that can help prevent the process through the removal of reactive oxygen species [https://doi.org/10.1016/bs.vh.2022.09.002]. We guess, that increasing of reduced glutathione aimed to protect cells from oxidative stress in stress condition. It is possible that the synthesis of glutathione in tissues is enhanced, which requires additional research.

Point 4:
The counterintuitive aspect of diabetic model data presented in Table 4 and fasting blood insulin (Figure 1f) need to be addressed. Specifically, Table 4 shows T2D group having significantly less number if islets per surface area (2nd row down) and percent insulin producing cells (4th row down) while Figure 1f shows no change in fasting blood insulin. Why isn’t there significant differences in fasting insulin values if BOTH the number of islets and the percent of insulin positive cells per islet are less? The authors need to address this question.

Response 4: It is known that β-cells dysfunction occurs before the manifestation of overt diabetes. May be, the decrease in both the number of islets and the percent of insulin positive cells per islet, may compensate by increase in the functional activity of β-cells, which allow to maintain the fasting blood insulin values in the 30th day of experimental diabetes. Moreover, besides islet β-cells, insulin may be produced by extra-islet insulin positive cells, localized in pancreatic acini and ducts. In current work we didn’t evaluate the functional activity of insulin positive cells and the mass of extra-islet insulin positive cells, but in our previous work was shown, that extra-islet IPCs mass decreased in experimental T2D less than islet β-cells mass [https://doi.org/10.3390/ijms23084286].

Failure of β-cell compensation, expressed in decrease both the number of islets and the percent of insulin positive cells per islet, leads to insulin resistance, which inputs in pathogenesis of T2D. HOMA-IR index in the 30th day of experimental diabetes demonstrated slight decrease without significant difference (Figure 1-g), but results of oral glucose tolerance test demonstrated significant difference with healthy state (Figure 1-d), which testify impaired glucose tolerance in rats with experimental diabetes. Impaired glucose tolerance is the intermediate state between normal glucose tolerance and overt diabetes.

Reviewer 2 Report

Comments and Suggestions for Authors

In the present study, the authors examined the protective effect of betulin on  Streptozotocin-Nicotinamide-induced diabetes in Rats. They observed that treatment with betulin improved the glycemic response and decreased α-amylase activity, hepatic biomarkers (AST, ALT, and ALP), and renal biomarkers (urea and creatine) in addition to improving glutathione levels and preventing the elevation of lipid peroxidation in diabetic animals

-The authors of this manuscript have employed streptozotocin (STZ) to induce diabetes in rats.  Does STZ administration result in the induction of type 1 or type 2 diabetes?   Please clarify this point.

-How did the authors arrive at the number of 10 rats/group? Was any statistical power analysis conducted to determine the rat number? If so, please include pertinent details.

-Should discuss the reason for selecting the betulin of doses of 20 and 50 mg/kg. Please justify. Also, indicate the margin of safety.

-Authors should give details of catalog numbers, and vendors of the chemicals, kits, and antibodies used.

-What anesthesia was used? What were the dosages of anesthesia? Please write sample collection details.

-Pancreatic tissues- How sections were prepared?

-Figure 3. Scale bar missing

-In the oxidative stress and Hepatic and renal function parameters, itemize and discuss each methodology within 2-3 lines and appropriate references.

-Statistical significances were missing in Figure 1e-g. Statistical analysis should be written more carefully in all tables and figures.

-In the discussion section, insert one paragraph and highlight the structure-activity relationship of betulin in ameliorating diabetes.

-More recent literature was added to the discussion, as suggested.

-The manuscript needs a very careful revision of the English language

Comments on the Quality of English Language

-The manuscript needs a very careful revision of the English language

Author Response

With respect to the comments from the Reviewer â„– 2:

Point 1:
The authors of this manuscript have employed streptozotocin (STZ) to induce diabetes in rats. Does STZ administration result in the induction of type 1 or type 2 diabetes?   Please clarify this point.

Response 1: There are a lot of rodent models of diabetes. Streptozotocin (STZ) is a diabetogenic agent, which damages pancreatic β cells, resulting in hypoinsulinemia and hyperglycemia [doi: 10.1002/0471141755.ph0547s70; DOI: 10.1358/mf.2009.31.4.1362513]. Standalone, Stz is widely used experimentally to produce a model of type 1 diabetes mellitus (T1DM) [DOI: 10.1002/0471141755.ph0547s70]. However, there are several protocols for modelling of T2D using Stz [https://doi.org/10.1002/cpz1.78]. We used concurrent administration of nicotinamide to partially protect the β-cells against STZ, first described by Marsiello [doi: 10.2337/diab.47.2.224]. Furthermore, the levels of hyperglycemia reproduced by the administration of Stz may be characteristic of both the first and second types of diabetes, depending on the type, age and weight of the animals, the nutritional status of the animal at the time of administration, the dose and route of administration, and susceptibility to xenobiotics [Arias-Díaz, J. Animal models in glucose intolerance and type-2 diabetes / J. Arias-Díaz, J. Balibrea // NutrHosp. – 2007. – Mar-Apr., vol. 22 (2). – P. 160-8; https://doi.org/10.1016/j.tice.2010.12.002].

Point 2:
How did the authors arrive at the number of 10 rats/group? Was any statistical power analysis conducted to determine the rat number? If so, please include pertinent details.

Response 2: We do not conduct statistical power analysis. To determine sample size, we use this method, named «tradition» [https://doi.org/10.1177/0023677217738268], since it is a common number in sample size in animal experimentation.

Point 3:
Should discuss the reason for selecting the betulin of doses of 20 and 50 mg/kg. Please justify. Also, indicate the margin of safety.

Response 3: Literature-based analysis showed, that betulin has been used as a treatment in both Stz- and alloxan-induced diabetes in rats in doses ranging from 20 to 100 mg/kg [doi: 10.1016/j.neuro.2016.09.009; doi: 10.32383/appdr/172620; doi: 10.1101/2023.07.27.550802]. Aside from these studies, other studies have demonstrated the broad pharmacological activity of betulin and categorized it as a nontoxic compound with a minimal lethal (LD16) and median lethal dose (LD50) of 6500 and 9000 mg/kg respectively [doi: 10.1556/2060.106.2019.26; doi: 10.1134/S1068162014070073; doi: 10.1016/j.intimp.2013.06.012]. The subchronic toxicity study of triterpene from the extract of the outer bark of birch, which includes betulin, showed no toxicity when scheduled at 300 mg/kg in 28 days [doi: 10.3390/molecules13123224].

Point 4:
Authors should give details of catalog numbers, and vendors of the chemicals, kits, and antibodies used.

Response 4: Thank you for your comment. All the required information for chemicals, kits and antibodies were incorporated in the revised version throughout the manuscript.

Point 5:
What anesthesia was used? What were the dosages of anesthesia? Please write sample collection details.

Response 5: Thank you for your comment. The required information was incorporated in sections 4.4, and 4.5 in which the following paragraphs were added:

“Prior studies have shown that betulin at a concentration of 20 and 50 mg/kg can signifi-cantly exert hypoglycemic effects on diabetic rats, hence, betulin emulsions was prepared and administered to rats by intragastric injection. [21,52]. All the intervention groups were treated for 28 days, thrice weekly. At the end of the experiment, the rats were deeply anesthetized with an i.m. injection of 0.1 mg/kg b.w. xylazine (Alfasan,Woerden, Nether-lands) and euthanized by i.m. injection of 5 mg/kg b.w. Zoletil-100 (Virbac, Carros, France).”

“Blood samples were collected before euthanasia from the rats via the tail vein and centrifuged”

“using the Novoglyuk-KM kit, catalog No 8039 (Vector-Best, Russian Federation)”

Point 6:
Pancreatic tissues - How sections were prepared?

Response 6: The sections for pancreatic tissues were prepared as was stated in the section 4.10:

“A median laparotomy was conducted to retrieve the pancreas. Pancreatic tissues were excised and fixed in a 10% formalin solution for 24 hours at room temperature. The tissues were then prepared for standard histological analysis and embedded in paraffin blocks using an automated processor (Leica EG 1160). Tissue sections with a thickness of 3–4 mm were prepared using sliding microtome Leica SM 2000R. An immunohistochemical procedure was performed using the avidin-biotin peroxidase complex (ABC) method. To detect insulin-positive cells, 4 μm thick pancreas paraffin sections were deparaffinized, de-hydrated, and washed in phosphate-buffered saline (3×; pH 6.0) at room temperature for 5 min. The pancreatic tissues were subjected to overnight incubation at 4 °C with an-ti-insulin and pro-insulin + insulin antibodies (clone INS04 + INS05, MA5-12042; Invitro-gen, Carlsbad, CA, USA) diluted at 1:200. Following incubation with a biotinylated secondary antibody for 1 h at 37 °C, a diaminobenzidine (DAB)-nickel reaction was conduct-ed, and all tissue sections were counterstained using hematoxylin and viewed under a microscope (Leica DM 2500) at an objective magnification of 40×.”

Point 7:
Figure 3. Scale bar missing.

Response 7: Thank you for your comment. We have incorporated scale bar to Figure 3 in the revised version of the manuscript.

Point 8:
In the oxidative stress and Hepatic and renal function parameters, itemize and discuss each methodology within 2-3 lines and appropriate references.

Response 8: In both cases newly information were added. In concrete:

For the case of oxidative stress:

“3 ml of 0.1 M phosphate buffer pH 7.4 containing 1% hydrogen peroxide and 10 μl of hemolysate were incubated for 10 minutes at 37ËšC, and the reaction was stopped by adding ammonium molybdate. To confirm the concentration of hydrogen peroxide solution at 0.3%, it was titrated against potassium permanganate and sulfuric acid (standard titers from EKROSHIN LLC, Russian Federation).

The determination of thiols and reduced glutathione was based on the reaction of -SH groups with 5,5′-dithiobis(2-nitrobenzoic) acid (DTNB, Abcam, UK) or Ellman's reagent at pH 8.0 [58]. For this assay, the supernatant obtained after precipitation of plasma proteins with 5.7% trichloroacetic acid in 0.25 M HCl (JSC Vekton, Russian Federation) was used and centrifuged at 1000 g for 10 minutes.

Malondialdehyde (MDA) is a by-product of free radical oxidation (FRO) that occurs after the formation of earlier products - hydroperoxides and diene conjugates, which are then converted to MDA and other aldehydes. Malondialdehyde and other FRO products containing aldehyde groups form compounds with thiobarbituric acid (TBA, Diaem, Russian Federation) in an acidic environment when heated. The optical density measured at a wavelength of 532 nm is directly proportional to the MDA concentration [55].

4.8. Hepatic and renal function parameters”.

For the case of hepatic and renal function parameters:

“AST and ALT enter the bloodstream when the integrity of the membrane is compromised, so AST and ALT activity are traditionally used to assess damaging processes in the liver, myocardium, and other tissues. An elevated AST/ALT ratio indicates prevalent myocardial injury, whereas a reduced AST/ALT ratio indicates liver injury.

   Alpha-amylase activity increases in blood plasma when the exocrine pancreas is damaged, and this enzyme is also a target for antidiabetic compounds since a decrease in its activity helps reduce glucose consumption [60]. In diabetic nephropathy, there is an elevation in the levels of urea and creatinine, while the levels of total protein decrease [61].

The plasma levels of AST (catalog No B 00.102), ALT (catalog No B 00.102), ALP (catalog No B 09.02), total protein (catalog No B 06.01), urea (B 08,02), α-amylase (catalog No B 11.01), and creatinine (catalog No B 04.02) were evaluated using kits provided by Vital Diagnostics (Russian Federation). Subsequently, a spectrophotometer (DU-800 Beckman Coulter Int. S.A., Nyon, Switzerland) was used to measure the absorbance at a specified wavelength following the instructions provided by the manufacturers.”

Point 9:
Statistical significances were missing in Figure 1e-g. Statistical analysis should be written more carefully in all tables and figures.

Response 9: The changes between the groups on these plots were not statistically different, hence the reason behind the absence of letters denoting significant differences.

Point 10:
In the discussion section, insert one paragraph and highlight the structure-activity relationship of betulin in ameliorating diabetes.

Response 10: We have done as recommended by including an additional paragraph at the end of the discussion

Point 11:
More recent literature was added to the discussion, as suggested.

Response 11: In the revised version of the manuscript, it was added 8 new references to the references list.

Point 12:
The manuscript needs a very careful revision of the English language.

Response 12: We have done a careful revision of our work to improve the language.

Reviewer 3 Report

Comments and Suggestions for Authors

This manuscript investigated the potential role of betulin, a triterpenoid natural compound, in protecting female rats from the damages caused by a model of type 2 diabetes mellitus (T2DM) induced by the association of streptozotocin and nicotinamide. The study and the search for new treatment candidates to mitigate T2DM effects on health are relevant considering the worldwide scenario of the disease. Nevertheless, the work could be improved by the following suggestions:

1) Abstract: Please rephrase the sentence “To assess the impact of betulin treatment on streptozotocin-nicotinamide-induced (STZ) diabetes in Wistar rats.” (Lines 18-19).

2) Why did the authors choose the doses of betulin (20 and 50 mg/Kg) and the treatment period (28 days)? It is not clear if these doses and the treatment period were based on prior literature or data from the research group.

3) Why the use of only female rats? The sex of the animals should also be disclosed in the title and abstract.

4) How was betulin obtained? Information on the obtention and preparation of betulin is partially missing and rather confusing. It is not clear if the betulin was purchased or if the authors extracted it from a natural source.

5) Also, is betulin a water-soluble compound? What was the vehicle for its administration? This information seems missing and should be included in the text.

6) As a suggestion, I believe that the results presented in Table 1 could also be moved to the supplementary material.

7) Line 98: Please provide an extension for FBG.

8) Just a curiosity: Did the authors evidence any signs of hypoglycemic shock after STZ administration?

9) Figure 1: It is not clear why some of the results presented in this figure do not contain all the experimental groups. Why were the other groups not investigated here?

10) The authors investigated some oxidative stress parameters in blood plasma. Did they think to evaluate these oxidative stress parameters in the pancreas?

11) Figure 3: Please use arrows to indicate significant changes in the immunohistochemical figures and include scale bars. The number of animals for histopathological assessment and other results should be also presented in the figure legends.

12) Line 264: Aspartate aminotransferase (AST) and alanine aminotransferase (ALT) are distinct enzymes. Please check and rephrase.

13) Line 311: Please provide an extension for AOD.

14) Lines 346-349: What about longer periods, what would the responses be? The duration of the study could be perhaps disclosed as a limitation.

15) Line 390: Please check and correct "years" to "grams".

16) Line 402: Why was this treatment schedule chosen and not every day, were there perhaps any signs of toxicity or unwanted side effects?

17) The manuscript could benefit from additional data, especially on the inflammatory pathways involved in the potential protective effect of betulin in the T2DM model used in this study. In the conclusion authors stated that “a change in hematological parameters indicates the occurrence of an inflammatory process” (Lines 481-482), but I believe that other assays (gene expression, western blot, …) should be performed to support these assumptions.

18) As a suggestion, the discussion could include some of the limitations of the study.

Author Response

Response to Reviewer â„– 3 Comments:

Point 1:
Abstract: Please rephrase the sentence “To assess the impact of betulin treatment on streptozotocin-nicotinamide-induced (STZ) diabetes in Wistar rats.” (Lines 18-19).

Response 1: The cited phrase was modified in the revised version of the manuscript according to Reviewer´s instructions.

Point 2:
Why did the authors choose the doses of betulin (20 and 50 mg/Kg) and the treatment period (28 days)? It is not clear if these doses and the treatment period were based on prior literature or data from the research group.

Response 2: Literature-based analysis showed, that betulin has been used as a treatment in both Stz- and alloxan-induced diabetes in rats in doses ranging from 20 to 100 mg/kg [doi: 10.1016/j.neuro.2016.09.009; doi: 10.32383/appdr/172620; doi: 10.1101/2023.07.27.550802]. Doses 10 mg/kg – 50 mg/kg demonstrated efficiency in the treatment of experimental diabetes. Aside from these studies, other studies have demonstrated the broad pharmacological activity of betulin and categorized it as a nontoxic compound with a minimal lethal (LD16) and median lethal dose (LD50) of 6500 and 9000 mg/kg respectively [doi: 10.1556/2060.106.2019.26; doi: 10.1134/S1068162014070073; doi: 10.1016/j.intimp.2013.06.012]. The subchronic toxicity study of triterpene from the extract of the outer bark of birch, which includes betulin, showed no toxicity when scheduled at 300 mg/kg in 28 days [doi: 10.3390/molecules13123224].

Point 3:
Why the use of only female rats? 2) The sex of the animals should also be disclosed in the title and abstract.

Response 3: We use animals of same sex (female) to reduce variability of the obtained data. We know that estrogen interferes with Stz action and female animals are less sensitive for diabetogenic action of Stz than the male rats [Kang HS, Yang H, Ahn C, Kang HY, Hong EJ, Jaung EB. Effects of xenoestrogens on streptozotocin-induced diabetic mice. J Physiol Pharmacol. 2014 Apr;65(2):273-82. PMID: 24781736], but we considered as diabetic and utilized for the study only animals with a blood glucose level ≥7.0 mmol/L and rejected all others.

Point 4:
How was betulin obtained? Information on the obtention and preparation of betulin is partially missing and rather confusing. It is not clear if the betulin was purchased or if the authors extracted it from a natural source.

Response 4: According to the suggestions from the Reviewer, in the revised version of the manuscript it was added the following information “Betulin obtained from Pro development Ltd (Yekaterinburg, Russia) was prepared into an emulsion as described by Zavorokhina et al. [46]»

Point 5:
Also, is betulin a water-soluble compound? What was the vehicle for its administration? This information seems missing and should be included in the text.

Response 5: Betulin has a low solubility in water (https://doi.org/10.3390/molecules28165946). Hence, we prepared betulin emulsion to facilitate oral administration as it was described in Section 4.1

Point 6:
As a suggestion, I believe that the results presented in Table 1 could also be moved to the supplementary material.

Response 6: According to the suggestion from the Reviewer, we moved the table to the supplementary materials.

Point 7:
Line 98: Please provide an extension for FBG.

Response 7: Thank you for your comment. In the revised version of the manuscript it was specified that “FBG” corresponds to “fasting blood glucose».

Point 8:
Just a curiosity: Did the authors evidence any signs of hypoglycemic shock after STZ administration?

Response 8: Stz selectively destroys β-cells, so it is widely used in biological and medical research to produce an animal model of diabetes. We used the concurrent administration of nicotinamide to partially protect the β-cells against STZ, this approach to modelling of diabetes first described by Marsiello [doi: 10.2337/diab.47.2.224].

Stz is a highly selected pancreatic β-cells-cytotoxic agent, it is known, that administration a single high dose of Stz cause complete β-cells necrosis and diabetes [doi: 10.1002/cpz1.78]. So, Stz can‘t lead to hypoglycemic shock.

Point 9:
Figure 1: It is not clear why some of the results presented in this figure do not contain all the experimental groups. Why were the other groups not investigated here?

Response 9: We injected betulin to non-diabetic healthy animals only in the dose 50 mg/kg. We suppose that the investigation effects of the larger dose of the testing substances (betulin) will be enough to evaluate its possible toxic effect.

Point 10:
The authors investigated some oxidative stress parameters in blood plasma. Did they think to evaluate these oxidative stress parameters in the pancreas?

Response 10: Yes, of course, we thought about it. However, we are faced with a methodology problem, concerning preparation of homogenates for biochemical analysis. The rat pancreas is a diffuse gland without the whole firm capsule, and it is rather difficult to separate pancreatic tissue from surrounding adipose tissue. So, during homogenization of pancreatic tissue homogenates may be contaminated with adipose drops.

Point 11:
Figure 3: (1) Please use arrows to indicate significant changes in the immunohistochemical figures and include scale bars. (2) The number of animals for histopathological assessment and other results should be also presented in the figure legends.

Response 11: According to the suggestion from the Reviewer, the cited information was included in the figure legends.

Point 12
Line 264: Aspartate aminotransferase (AST) and alanine aminotransferase (ALT) are distinct enzymes. Please check and rephrase.

Response 12: We agree with the point raised and have corrected it

Point 13:
Line 311: Please provide an extension for AOD.

Response 13: Thank you for your comment. In the revised version of the manuscript it was specified that “AOD” corresponds to “antioxidant defense».

Point 14
Lines 346-349: What about longer periods, what would the responses be? The duration of the study could be perhaps disclosed as a limitation.

Response 14: In the current work we studied effect of betulin treatment only at the early stages of the experimental diabetes. It was included in the newly included section of limitations of the study that the duration of study could be a limitation.

Point 15:
Line 390: Please check and correct "years" to "grams".

Response 15: Thank you for your comment. This mistake was corrected in the revised version of the manuscript.

Point 16:
Line 402: Why was this treatment schedule chosen and not every day, were there perhaps any signs of toxicity or unwanted side effects?

Response 16: We avoided daily treatment to reduce stress for animals from intragastric injections.

Point 17:
The manuscript could benefit from additional data, especially on the inflammatory pathways involved in the potential protective effect of betulin in the T2DM model used in this study. In the conclusion authors stated that “a change in hematological parameters indicates the occurrence of an inflammatory process” (Lines 481-482), but I believe that other assays (gene expression, western blot, …) should be performed to support these assumptions.

Response 17: In current work ability of betulin to reduce hyperglycemia and oxidative stress and absence of toxicity were shown, but we did not investigate effects of betulin treatment at the lipid metabolism and the inflammatory level, which both involves in the pathogenesis of diabetes. This may be attributed to the limitations of the work.

Point 18:
As a suggestion, the discussion could include some of the limitations of the study.

Response 18. According to your suggestion, a new paragraph of limitations of the study was added to the revised version of the manuscript.

Round 2

Reviewer 1 Report

Comments and Suggestions for Authors

The authors did a good job of addressing all reviewers' comments.